# Currency risk analysis of belt and road initiative countries

**Mengyang Li**[1]*, **Wencui Liu**[1], **Lei Wang**[2,3]*

**1** School of Finance, Xinjiang University of Finance & Economics, Urumqi, China, **2** State Key Laboratory of Desert and Oasis Ecology, Xinjiang Institute of Ecology and Geography, Chinese Academy of Sciences, Urumqi, China, **3** University of Chinese Academy of Sciences, Beijing, China

* lmy-x@hotmail.com (ML); egiwang@ms.xjb.ac.cn (LW)

**Data Availability Statement:** All relevant data are within the manuscript and its Supporting Information files.

## Abstract

The UN's Sustainable Development Goals (SDGs) highlight the role of debt sustainability in achieving sustainable development. China's Belt and Road Initiative (BRI) is an international cooperation effort that is endorsed by over 150 countries and organizations. Given the alignment between BRI development goals and the SDGs, the issue of debt sustainability in BRI countries warrants attention. While existing studies focus on sovereign risk in debt sustainability, there is a lack of emphasis on currency risk, indicating a need for further investigation to mitigate risks and comprehensively evaluate debt stability. Using data from 142 countries, this study examines currency risk reduction in BRI countries by assessing currency competitiveness. We find that the US dollar (USD) is the most competitive currency among BRI countries, followed by the Euro (EUR), Chinese yuan (CNY), sterling pound (GBP), and Japanese yen (JPY). The USD maintains its competitive edge over time, making it the preferred choice, with the EUR as a less optimal option and the CNY showing potential. Geographically, the EUR's close ties with BRI countries lend it prominence, followed by the USD, with the CNY gaining traction. GBP and JPY are considered conservative choices. Recommendations for currency selection vary based on countries' competitiveness, bilateral relationships, and development status.

## 1. Introduction

In October 2015, the United Nations (UN) released *Transforming Our World: The 2030 Agenda for Sustainable Development*, outlining 17 Sustainable Development Goals (SDGs) focused on eradicating poverty and promoting economic prosperity. A couple of years earlier, in 2013, China had introduced its Belt and Road Initiative (BRI), which was inspired by the historical Silk Road. The BRI, detailed in "The Belt and Road Initiative: A Key Pillar of the Global Community of Shared Future (2023)," released by The State Council Information Office, aims to enhance the sustainability and well-being of its participants. A comparison of the SDGs and the BRI reveals alignment in key areas, emphasizing sustainable development and economic growth for all of the involved nations [1–6].

Debt sustainability is critical for BRI countries, especially developing nations that face challenges such as inadequate infrastructure and limited capital, which hinder economic growth

**Funding:** the State Key Laboratory of Desert and Oasis Ecology: Xinjiang Institute of Ecology and Geography: CAS (E1510107).

**Competing interests:** The authors have declared that no competing interests exist.

and foreign investment [7]. Even countries with development potential face debt sustainability risks, exacerbated by rising debt levels among developing nations, as highlighted in the UN's "Sustainable Development Goals Report 2023." Addressing such risks is essential for bolstering financial resilience in BRI countries. The International Monetary Fund (IMF) and the World Bank's Debt Sustainability Framework (DSF) track countries' debt capacity, focusing on factors such as public and public-guaranteed external debt, GDP, and revenue. According to the DSF, a country's debt risk encompasses sovereign credit risk, default risk, and liquidity risk, all of which can negatively affect a debtor country's ability to service its debt [8]. While many studies have examined the default risk and sovereign risk of debtor countries [9,10], such research often overlooks the importance of currency risk. International transactions, influenced by exchange rates and diverse currencies, introduce uncertainties and potential losses, commonly referred to as currency risk or exchange-rate risk [11,12]. This study shifts the focus to currency risk and explores strategies by which debtor countries can better manage it. Developing countries, including the majority of BRI nations, can effectively manage external debt by strategically selecting currency combinations to minimize currency risk. This requires carefully navigating currency risk by selecting optimal currency combinations to protect their financial resources and ensure debt sustainability amid the complexities of international transactions.

This study is closely related to the literature on currency risk, which mainly develops in two directions: the currency risk of domestic currency–denominated external debt in the international market and the currency risk of foreign currency–denominated external debt. The former, also known as "original sin," describes the challenge governments face in raising debt in their domestic currency [13]. Many studies have investigated this, exploring how factors such as inflation costs and exchange-rate stability affect original sin using dynamic stochastic general equilibrium (DSGE) and general equilibrium models, among others, where government commitment to debt and monetary policy is lacking [14,15]. Using domestic currency for invoicing is seen as a potential strategy for mitigating the currency risk of debtors in developing countries, thereby improving their debt sustainability [16–18]. Challenges persist, however, owing to the vulnerability of domestic currencies in open markets and the strong preference for developed-country currencies or the US dollar (USD) in international transactions [19–24]. Given that original sin remains a persistent challenge [25], especially for developing countries, it is crucial to also examine foreign currency–denominated external debt.

The other direction in the currency risk literature considers the risks associated with foreign currency denominations in external debt. In the 1990s, Claessens highlighted the importance of the currency composition of external debt and introduced optimal portfolio theory to help optimize it [26,27]. Subsequent researchers developed various models and conducted empirical analyses. For instance, using a model focused on foreign debt portfolio management, one study found that exchange-rate fluctuations significantly affect currency composition, yet developing countries often fail to adjust appropriately when the USD appreciates [28]. Furthermore, the use of the cost-at-risk—a risk index for debt–service–cost risk—led to the creation of a benchmarking framework for government debt based on a penalty function [29]. Studies of low-income countries have revealed that currency concentration exacerbates exchange-rate procyclicality, leading to greater volatility in domestic consumption during periods of national income volatility. Diversifying the currency composition of external debt has been shown to potentially mitigate these negative effects [30]. While some studies examine currency composition using behavioral portfolio theory and a capital asset pricing model (CAPM), these perspectives are more creditor oriented than debtor oriented [31,32]. Overall, there is general consensus that an optimal currency composition can effectively reduce currency risk [33].

Existing currency risk analyses often overlook the unique nature and characteristics of international currencies, neglecting their special functions and hierarchical competition. Common approaches, such as the DSF, currency-mismatch, and portfolio approaches, have various limitations. For instance, while DSF can thoroughly evaluate debt sustainability and risk [34–37], it tends to place less emphasis on currency risk. Meanwhile, the currency-mismatch indicator focuses on maturity-length mismatch without considering currency composition [38–40]. Similarly, the portfolio approach focuses on currency fluctuations and exchange-rate premiums [30] but fails to address the crucial aspects of international currencies.

To address these limitations, this study uses currency competition theory to analyze currency risk for debtor countries, a framework that considers the unique characteristics of international currencies. Hayek's *Denationalization of Money* and Cohen's currency pyramid model lay a foundation for understanding currency competition [41,42]. Economists such as McKinnon and Tavlas highlighted the interplay among invoicing practices, the reserve capacity of a currency, and the standing of the currency-issuing country in international trade and finance [20,21,43]. Furthermore, metrics such as the currency power index and currency internationalization index offer multidimensional assessments of currency influence [44–47]. Based on the theory of optimal currency areas, the anchor effect of currency highlights the regional effects of currencies and their role in mitigating exchange-rate risk in international trade [48]. We posit that currency competitiveness rests on three pillars: currency stability, currency invoicing capacity, and currency reserve capacity.

In managing currency risk, the competitiveness of foreign currencies plays a pivotal role, especially in the bilateral relationship between the debtor country and the currency-issuing country. Opting for competitive currencies benefits debtor countries by signaling stability and garnering wider acceptance [41], thereby reducing risk. Highly competitive currencies also benefit from strong network externalities and inertia, rendering them more resilient to risk [49,50], especially in challenging economic climates [51,52]. Additionally, countries with close bilateral ties face less risk when transacting in each other's currencies, since factors such as spatial proximity, bilateral trade volume, and investment levels influence currency selection. By considering both currency competitiveness and bilateral relationships, debtor countries can make well-informed decisions about optimal currency selection to effectively manage their currency risk.

This study evaluates the performance of influential currencies in BRI countries and offers insights into optimal currency-selection strategies to bolster debt sustainability and mitigate risk. To identify suitable currency options for BRI countries, we apply the theory of currency competition and establish a currency competitiveness index (CCI), following the method outlined in the OECD's *Handbook on Constructing Composite Indicators*: *Methodology and User Guide* [53,54]. Building on Hayek, who defines currency competitiveness in terms of stability, invoicing capacity, and reserve capacity, we examine spatiotemporal considerations in currency selection and deliver tailored recommendations for currency strategies. In what follows, we present the CCI measurement method, describe the CCI results, further discuss optimal currency selection for BRI countries, and draw conclusions.

## 2. Methods

### 2.1 CCI measurement system and indicators

CCI is structured around three pillars, each supported by various indicators. Pillar I, currency stability, includes three tier 2 indicators that describe stability in three dimensions—military, political, and economic—related to the currency-issuing country's situation. Pillar II, currency invoicing capacity, comprises two tier 2 indicators: currency invoicing capacity on

international trading and in international finance. Pillar III, currency reserve capacity, incorporates one tier 2 indicator: international reserve capacity.

To capture each tier 2 indicator, we select tier 3 indicators based on previous studies. For the tier 3 indicators of pillar I, the percentage of military expenditure (MEGDP) and the level of political stability (i.e., political stability and the absence of violence/terrorism (PSAV)) are commonly used to characterize military stability and political stability, respectively [55,56]. Economic stability is often represented by exchange rate volatility (VOLFX) and inflation (DINF), which also reflect the external and internal value stability of the currency [57].

For the tier 3 indicators of pillar II, the percentage of exports (EXALL), percentage of trade with developing countries (EXEDE), and percentage of trade in intermediate products (EXINTER) reflect currency influence in BRI countries. Tavlas noted that the share of a currency-issuing country's trade with other countries, trade with developing countries, and trade in intermediate goods reflect the local influence of the country's currency [20]. These export share indicators have been used in studies of international trading [58]. Therefore, we introduce export share indicators into the currency-based index. Additionally, the percentage of outward foreign direct investment (ODI), capital account openness (KAOPEN), the percentage of debt securities issued (ISSU), and the percentage of outstanding debt securities (OUTS) are commonly used when studying currency invoicing capacity in international financial markets [59,60]. Following Xu [61], we also consider the anchor effect of currency (ANC). ANC signifies the exchange rate effect of a country's currency in a specific region. We use a time-varying parameter state-space model developed by Zhou [62], based on the classic ANC model of Frankel and Wei (Eq 1) [48], to capture the evolving trend of a currency's weight in the currency basket of BRI countries.

$$\Delta \ln Y_t = c + \sum \omega(j)[\Delta \ln X_{j,t}] + \varepsilon_t, \tag{1}$$

where $\Delta \ln Y_t$ denotes the first-order difference in the natural logarithm of the exchange rates of the currencies of BRI countries against Special Drawing Rights (SDR) in period $t$; $\omega(j)$ represents the weight of currency $j$ in the basket of currencies of BRI countries, known as ANC; $\Delta \ln X_{j,t}$ denotes the first-order difference in the natural logarithm of the exchange rate of currency $j$ against SDRs; $c$ signifies the constant term; and $\varepsilon_t$ accounts for the random error term. This model indicates that each exchange rate of BRI countries is influenced by other currencies to varying degrees. $\omega(j)$ in the model captures the extent to which the exchange rate of a BRI country is influenced by currency $j$, reflecting the structure of the currency basket of the BRI country.

Before measuring $\omega(j)$, a preliminary issue must be addressed. When applying the classic ANC model to the five currencies, the Chinese yuan (CNY) exchange rate cannot be directly used. Given that the Chinese capital market is partially closed to foreign capital, the international status of the CNY is somewhat limited. The CNY exchange rate is affected by a range of currencies, including the USD, the euro (EUR), the Japanese yen (JPY), and the British pound sterling (GBP), among others. To separate the fluctuations of the CNY exchange rate from those of other currencies and focus on its inherent trend, we conduct a regression analysis of the CNY exchange rate. The residuals $\varepsilon_{CNY,t}$ derived from this regression represent the component of CNY exchange rate changes (Eq 2), which are then employed in the ANC model:

$$\Delta \ln CNY_t = c + \alpha_{USD,t}\Delta \ln USD_t + \alpha_{EUR,t}\Delta \ln EUR_t + \alpha_{JPY,t}\Delta \ln JPY_t + \alpha_{GBP,t}\Delta \ln GBP_t$$
$$+ \varepsilon_{CNY,t}, \tag{2}$$

where $\varepsilon_{CNY,t}$ represents the independent movement of the CNY exchange rate. With this issue addressed, we successfully isolate fluctuations in the CNY exchange rate. Let

$\varepsilon_{CNY,t} = \Delta \ln \hat{CNY}_t$, and adopt a time-varying parametric model to derive the ANC model (Eq 3):

$$\Delta \ln Y_{i,t} = c + \beta_{USD,i,t}\Delta \ln USD_t + \beta_{EUR,i,t}\Delta \ln EUR_t + \beta_{JPY,i,t}\Delta \ln JPY_t + \beta_{GBP,i,t}\Delta \ln GBP_t$$
$$+ \beta_{CNY,i,t}\Delta \ln \hat{CNY}_t + \varepsilon_t, \tag{3}$$

where the regression coefficient $\beta_{j,i,t}$ represents ANC $j$, with higher values indicating a more significant effect of currency $j$ in the BRI area. This implies that BRI countries are more likely to include currency $j$ as a member currency in their exchange rate basket, leading to a closer relationship between BRI countries and the currency-issuing country. The regression coefficient $\beta_{j,i,t}$ serves as the tier 3 indicator ANC.

Regarding the tier 3 indicators of pillar III, the percentage of international reserves (COFER) is often used as an indicator in the literature and is used here as well [60,61]. Hence, in this study, CCI comprises a total of 13 tier 3 indicators, offering a more comprehensive assessment of currency competitiveness compared with previous studies (Table 1).

**Table 1. Currency competitiveness index measurement system.**

| Index | Tier 1 Indicator | Tier 2 Indicator | Tier 3 Indicator | Tier 3 Indicator Code | Description of Tier 3 Indicator |
|---|---|---|---|---|---|
| Currency Competitiveness Index (CCI) | Pillar I Currency Stability | Military Stability | Percentage of Military Expenditure | MEGDP | Measures the currency-issuing country's military expenditure as a percentage of its GDP |
| | | Political Stability | Level of Political Stability | PSAV | Uses the indicator political stability and the absence of violence/terrorism from the WGI, assessing perceptions of the likelihood of political instability, violence, and terrorism |
| | | Economic Stability | Exchange Rate Volatility | VOLFX | Evaluates monthly exchange rate volatility |
| | | | Rate of Change in Inflation | DINF | Measures the year-on-year inflation rate of change |
| | Pillar II Currency Invoicing Capacity | Currency Invoicing Capacity on International Trading | Percentage of Export | EXALL | Represents the currency-issuing country's exports as a share of total global exports to BRI countries |
| | | | Percentage of Developing Countries | EXEDE | Indicates the currency-issuing country's exports as a share of global exports to developing countries among the BRI countries |
| | | | Percentage of Intermediate Products | EXINTER | Reflects the currency-issuing country's exports as a share of global exports of international commodities to BRI countries |
| | | Currency Invoicing Capacity on International Finance | Percentage of Outward FDI | ODI | Measures the currency-issuing country's outward FDI as a share of global outward FDI flows |
| | | | Capital Account Openness | KAOPEN | Assesses a country's level of capital account openness [63] |
| | | | Anchor Effect of Currency | ANC | Represents coefficients corresponding to each currency in the regression results of the anchor effect of currency |
| | | | Percentage of Debt Securities Issues | ISSU | Indicates the proportion of global international bond and note issuance in bonds and notes by currency |
| | | | Percentage of Debt Securities Amounts Outstanding | OUTS | Represents the share of global international bond and note balances in bonds and notes by currency |
| | Pillar III Currency Reserve Capacity | International Reserve Capacity | Percentage of International Reserves | COFER | Measures the reserve share of global foreign exchange reserves by currency, reflecting the level of official recognition of varying currency reserve capacities across countries |

## 2.2 Data and processing

**2.2.1 Sample and data sources.** We use quarterly data from 1996 to 2020 for the USD, EUR, JPY, GBP, and CNY at the currency level. As of January 31, 2021, 140 countries had entered into partnership agreements with China for BRI development; eight countries in the BRI region have not yet signed such agreements. To ensure the reliability and comprehensiveness of the data, we select a sample of 142 BRI countries, consisting of 10 countries in North America, nine in Oceania, 46 in Africa, eight in South America, 27 in Europe, and 42 in Asia. Table 2 presents the sample size, time frame, frequency, and data origins for the tier 3 indicators.

**2.2.2 Data processing.** To deal with missing values and low-frequency data, we use the mean interpolation method and quadratic matched averaging method for data transformation. In the original database, certain variables are initially recorded as monthly or yearly data and then converted into quarterly data. The conversion process entails transforming annual,

**Table 2. Sample status and data source of tier 3 indicators.**

| Tier 3 Indicator Code | Period | Frequency | Sample Size | Number of Unestimated Observations | Missing Data | Data Source |
|---|---|---|---|---|---|---|
| MEGDP | 1996–2020 | Yearly | 25 | 575 | 0 | SIPRI Military Expenditure Database |
| PSAV | 1996–2020 | Yearly | 25 | 506 | 0 | WB-WGI |
| VOLFX | 1996–2020 | Monthly | 300 | 1500 | 0 | US Federal Reserve |
| DINF-USD | 1996–2020 | Monthly | 300 | 300 | 0 | Bureau of Labor Statistics of the US Department of Labor |
| DINF-EUR | 1996–2020 | Monthly | 288 | 288 | 0 | European Central Bank |
| DINF-JPY | 1996–2020 | Monthly | 300 | 300 | 0 | Statistics Bureau of the Ministry of Internal Affairs and Communications of Japan |
| DINF-UK | 1996–2020 | Monthly | 300 | 300 | 0 | Office for National Statistics of the UK |
| DINF-CNY | 1996–2020 | Monthly | 300 | 300 | 0 | National Bureau of Statistics of the China |
| EXALL | 1995–2020 | Yearly | 3480 | 756470 | 7952 | UN Comtrade Database |
| EXEDE | 1995–2020 | Yearly | 3480 | 756470 | 7952 | UN Comtrade Database |
| EXINTER | 1995–2020 | Yearly | 3480 | 756470 | 7952 | UN Comtrade Database |
| ODI | 1996–2020 | Yearly | 25 | 114 | 8 | IMF-IFS |
| KAOPEN | 1996–2020 | Yearly | 25 | 550 | 0 | Chinn–Ito index |
| ANC-5 | 1996–2020 | Daily | 6380 | 29502 | 0 | IMF Exchange Rate Report |
| ANC-BRI | 1996–2020 | Monthly | 42600 | 41230 | 1370 | IMF-IFS |
| ISSU | 1996–2020 | Quarterly | 100 | 549 | 51 | BIS |
| OUTS | 1996–2020 | Quarterly | 100 | 600 | 0 | BIS |
| COFER | 1996–2020 | Quarterly | 100 | 369 | 0 | IMF-COFER |

monthly, and daily data into quarterly data. For instance, as shown in Table 2, yearly data such as MEGDP, PSAV, EXALL, EXEDE, EXINTER, ODI, and KAOPEN; monthly data such as VOLFX, DINF, ANC-BRI (i.e., foreign exchange rate data of BRI country currencies used for ANC); and daily data such as ANC-5 (i.e., foreign exchange rate data of five currencies used for ANC) are converted accordingly.

We use two main methods to address missing data for specific years in individual countries: the modeling prediction method and the regression difference method. For instance, variables such as EXALL, EXEDE, and EXINTER lack observations for 1996 and 1997, but data for 1995 are available. In such cases, the observations for 1996 are estimated using data from 1995, followed by the use of the mean interpolation method to estimate observations for 1997. Missing data for the variables ODI and ISSU from the early years of China and the Eurozone are estimated using time series models. Partial missing data for ANC-BRI are filled during the frequency conversion process while the remaining portion is processed using the mean interpolation method.

VOLFX is considered an inverse indicator of currency stability, with higher volatility values indicating greater currency instability. We use the common min-max normalization method to transform these inverse indicators into a positive direction. Post-transformation, higher VOLFX values correspond to enhanced currency stability. The min-max normalization equation is

$$mmx = \frac{x_i - x_{min}}{x_{max} - x_{min}}. \tag{4}$$

We use the Z-score method to standardize the data of 12 tier 3 indicators. However, as the capital account convertibility level represented by the KAOPEN index is already a composite index, it is exempt from Z-score processing. The equation for calculating Z-score is as follows:

$$z = \frac{x - \mu}{\sigma}. \tag{5}$$

## 2.3 Weighting and aggregation

**2.3.1 Weighting.**  We use the entropy weight method to allocate weights to the tier 3 indicators; this method is commonly used in index construction [60,61]. Rooted in information theory, entropy weight quantifies information disorder. By examining historical data, the entropy weight method assigns weights to each indicator based on the entropy value, which captures the information variance in each indicator. These weights are impartial, are free from individual bias, and effectively represent the influence of each indicator on the overall index.

The process entails constructing the initial evaluation matrix $X = (X_{ij})_{m \times n}$ using the data from the measurement system, where n represents the evaluation indicators, $m$ denotes the objects under evaluation, and $t$ signifies the time periods. This matrix is then normalized to generate the evaluation matrix $R = (r_{ij})_{m \times n}$. The weight $p_{ij}$ of indicator $j$ in period $i$ is calculated as follows:

$$p_{ij} = \frac{r_{ij}}{\sum_{i=1}^{m} r_{ij}}, \tag{6}$$

$$k = \frac{1}{\ln t}. \tag{7}$$

**Table 3. Entropy weights for tier 3 indicators of CCI.**

| Tier 3 Indicators | Indicator Code | Entropy Weights |
|---|---|---|
| Percentage of Military Expenditure | MEGDP | 0.0778 |
| Level of Political Stability | PSAV | 0.0848 |
| Exchange Rate Volatility | VOLFX | 0.1082 |
| Rate of Change in Inflation | DINF | 0.1057 |
| Percentage of Exports | EXALL | 0.0768 |
| Percentage of Developing Countries | EXEDE | 0.0773 |
| Percentage of Intermediate Products | EXINTER | 0.0761 |
| Percentage of Outward FDI | ODI | 0.0741 |
| Capital Account Openness | KAOPEN | 0.0104 |
| Anchor Effect of Currency | ANC | 0.0737 |
| Percentage of Debt Securities Issued | ISSU | 0.0793 |
| Percentage of Debt Securities Amounts Outstanding | OUTS | 0.0791 |
| Percentage of International Reserves | COFER | 0.0767 |

The information entropy $e_j$ of indicator $j$ is obtained by substituting $p_{ij}$ and $k$ into Eq (8):

$$e_j = -k\Sigma_{i=1}^{m} p_{ij} \times \ln p_{ij}. \tag{8}$$

The entropy weight coefficient $w_j$ for indicator $j$ is obtained using Eq (9):

$$w_j = \frac{(1 - e_j)}{\sum_{j}^{n}(1 - e_j)}. \tag{9}$$

Table 3 shows the entropy weight results derived from the tier 3 indicator data of CCI.

### 2.3.2 Aggregation

Following the steps outlined in the OECD Handbook, once the weights of the 13 tier 3 indicators are determined, CCI can be aggregated using Eq (10):

$$CCI = \sum_{j=1}^{k} w_j x_j, \tag{10}$$

where $w_j$ denotes the weight of indicator $j$ estimated by entropy weighting; $x_j$ denotes indicator $j$, and $k$ denotes the number of indicators.

### 2.4 Forecasting model

We have calculated CCI for five currencies. To further investigate and monitor trends in each currency's CCI, the next step involves forecasting CCI. The autoregressive integrated moving average (ARIMA) model is widely used to forecast time-series data because of its systematic modeling approach and reliable forecasting performance. We use ARIMA to predict CCI on out-of-sample data. Eq (11) presents the ARIMA (p, d, q) model:

$$\left(1 - \sum_{i=1}^{p} \phi_i L^i\right)(1 - L)^d CCI_t = \left(1 + \sum_{i=1}^{q} \Theta_i L^i\right)\varepsilon_t$$

$$d \in \mathbb{Z}, d > 0, \tag{11}$$

where $(1 - \sum_{i=1}^{p} \phi_i L^i)CCI_t$ represents the AR (p) part of ARIMA, $p$ is the number of autoregressive terms, $(1-L)^d CCI_t$ represents the I (d) part of ARIMA, $d$ is the order of differencing, $(1 + \sum_{i=1}^{q} \Theta_i L^i)\varepsilon_t$ represents the MA (q) part of ARIMA, $q$ is the number of moving average terms, and $L$ represents the lag operator.

Constructing an ARIMA model typically involves four phases: identification and order determination, parameter estimation, model testing, and forecasting. In the first phase, identification and order determination focus on establishing the parameters $p$, $d$, and $q$. Parameters $p$ and $q$ are initially determined using the autocorrelation function and partial autocorrelation function. Parameter $d$ represents the amount of differencing required to achieve stationarity, with the goal of stabilizing the data. Commonly used methods include the first-order and second-order difference methods, corresponding to $d$ values of 1 and 2, respectively. The second phase, parameter estimation, involves confirming the suitability of the ($p$, $d$, $q$) order and selecting appropriate values for $p$, $d$, and $q$ based on criteria such as the Akaike information criterion (AIC) or Bayesian information criterion. Additionally, we obtain parameter estimates for the ARIMA model. During the third phase, we conduct model testing to assess the significance of parameter estimates using a t-test and test the randomness of the residual sequence using the Ljung–Box Q test. Passing these tests ensures the model's validity for forecasting purposes. In the final phase, forecasting is performed based on the ARIMA model selected through the previous steps, allowing for the prediction of out-of-sample data.

## 3. Results

### 3.1 CCI of USD, EUR, JPY, GBP, and CNY

**3.1.1 CCI trend.** We focus on CCI after BRI establishment—specifically, the period after 2013Q3, when BRI was introduced. Here, we examine the CCI of five currencies from 2013Q3 to 2020Q4. The period 1996Q1–2013Q2 is used for further comparison and forecasting. Among BRI countries, we find variations in the CCI fluctuations of the USD, EUR, JPY, GBP, and CNY.

The competitiveness of the USD remains consistently high, with a mean value of 2.1301. Despite a minor decline in 2015Q4, USD competitiveness remains relatively stable from 2013Q3 to 2020Q4, consistently ranking first among the five currencies. The EUR's competitiveness closely tracks that of the USD, with a mean value of 2.0083. While initially comparable to USD competitiveness from 2013Q3 to 2014Q4, EUR competitiveness deviates around 2015, fluctuating around 1.80.

JPY ranks fifth in competitiveness, with a mean of 1.0318. During the Crimean crisis of 2014, the failure of the JPY's safe-haven currency status and Japan's quantitative easing policy resulted in a significant depreciation of the JPY and a widening trade deficit, leading to a sharp decline in competitiveness. GBP ranks fourth in competitiveness, with a mean of 1.3118, showing a slight decline. CNY ranks third in competitiveness, with a mean of 1.5805. The CNY's competitiveness has steadily increased since the inception of the BRI (Fig 1 and Table 4).

**3.1.2 CCI distribution.** The five currencies exert varying influences among BRI countries. For currencies with some or all international monetary functions—as with the five currencies examined here—their effect is felt by countries in close geographical proximity. Analyzing the geographical distribution of their competitiveness offers insights into which currency holds more sway over these nations. Taking 2020Q4 as a case study, we examine the spatial distribution of the competitiveness of the five currencies. The geographical perspective in this study is defined by continent affiliation. The 142 countries under consideration are categorized into

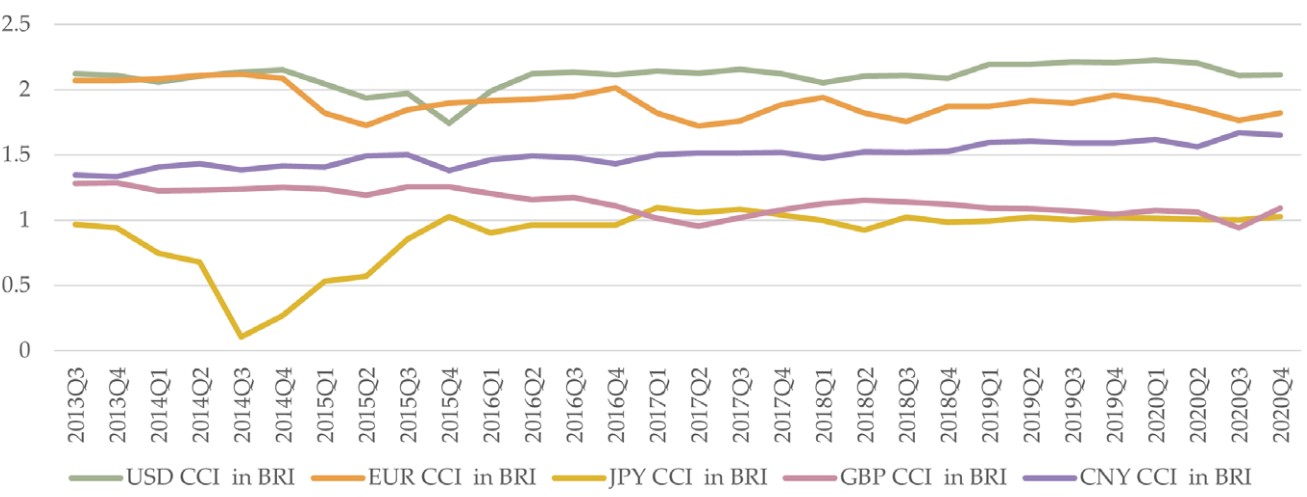

**Fig 1. CCI of five currencies in BRI countries.**

six groups based on continental affiliation—namely, Africa, Asia, Europe, North America, South America, and Oceania.

Among the top 20 countries in the CCI rankings for each currency, those with high CCI for the USD are concentrated in North and South America; for the EUR, in Europe and Africa; for the JPY, in Asia and Africa; for the GBP, in Africa and Asia; and for the CNY, in Africa and Asia (Table 5).

To further analyze the distribution of BRI countries based on each currency's CCI, we rank the CCI of the 142 countries for 2020Q4 from highest to lowest for each currency. For the USD, the most competitive country is Dominica in North America (CCI: 2.4409) while the least competitive is Comoros in Africa (CCI: 2.0590). About 110 countries, or 77.46%, fall below the average CCI of 2.1139 for the USD. In the case of the EUR, the most competitive country is Cabo Verde in Africa (CCI: 2.3150) while the least is Micronesia in Oceania (CCI: 1.8404). About 99 countries, or 69.72%, have CCIs below the average of 1.9395 for the EUR.

For the JPY, the most competitive country is Turkmenistan in Asia (CCI: 1.2921), with Botswana in Africa being the least competitive (CCI: 1.1582). About 103 countries, or 72.54%, have CCIs below the average of 1.1697 for the JPY. Senegal in Africa is the most competitive country for GBP (CCI: 1.5268) while Timor-Leste in Asia is the least competitive (CCI: 1.2097). There are 112 countries, or 78.87%, with CCI below the average of 1.2279 for the GBP. In the case of the CNY, Turkmenistan in Asia is the most competitive (CCI: 2.0574) while Luxembourg in Europe is the least competitive (CCI: 1.5942). About 77 countries, or 54.22%, have CCIs below the average of 1.7037 for the CNY.

**Table 4. Descriptive statistics of CCI.**

| CCI | Sample Size | Mean | Standard Deviation | Minimum | Maximum | Extreme Difference | Median |
|---|---|---|---|---|---|---|---|
| USD | 4260 | 2.1301 | 0.1163 | 1.7299 | 2.5676 | 0.8378 | 2.1405 |
| EUR | 4260 | 2.0083 | 0.1492 | 1.7074 | 2.5977 | 0.8903 | 1.9840 |
| JPY | 4260 | 1.0318 | 0.2363 | 0.2328 | 1.4069 | 1.1741 | 1.1300 |
| GBP | 4260 | 1.3118 | 0.0992 | 1.1018 | 1.6985 | 0.5967 | 1.3096 |
| CNY | 4260 | 1.5805 | 0.1049 | 1.3559 | 2.0749 | 0.7189 | 1.5732 |

**Table 5. Top 20 CCI countries by currency and continent in 2020Q4.**

|  | USD CCI in BRI | EUR CCI in BRI | JPY CCI in BRI | GBP CCI in BRI | CNY CCI in BRI |
|---|---|---|---|---|---|
| **Africa** | Zimbabwe, South Sudan | Cabo Verde, Tunisia, Morocco, Algeria, Gabon, Mali, Mauritania, Angola, Côte d'Ivoire | Somalia, Zimbabwe, South Sudan, Kenya, Liberia | Senegal, Zimbabwe, South Sudan, Angola, Ethiopia, Libya, South Africa | Liberia, Djibouti, Nigeria, Gambia, Ghana, Guinea, Togo, Benin, Madagascar, Somalia |
| **Asia** | Israel | Turkmenistan, Turkey | Turkmenistan, Thailand, Korea, Philippines, Indonesia, Malaysia, Vietnam, Singapore, Turkey, India, Kuwait, Mongolia | Turkmenistan, Qatar, Azerbaijan, Brunei Darussalam, United Arab Emirates, Turkey | Turkmenistan, Myanmar, Mongolia, Timor-Leste, Vietnam, Iran, Cambodia, Kyrgyzstan |
| **Europe** | Ukraine | Bosnia and Herzegovina, Serbia, Moldova, Albania, North Macedonia, Ukraine, Russian, Montenegro | Russian | North Macedonia, Belarus, Malta, Estonia, |  |
| **North America** | Dominica, Dominican Republic Costa Rica, Antigua and Barbuda, Trinidad and Tobago, Jamaica, Barbados, El Salvador, Grenada |  |  | Antigua and Barbuda, Barbados |  |
| **South America** | Guyana, Venezuela, Suriname, Ecuador, Chile, Peru, | Suriname |  |  |  |

The first four currencies exhibit higher CCIs for the more competitive countries, leading to a larger number of countries falling below the mean. Compared with the other four currencies, CNY displays a smaller difference between the extremes of its CCI, with fewer countries having high CCI, resulting in a relatively even distribution (Fig 2).

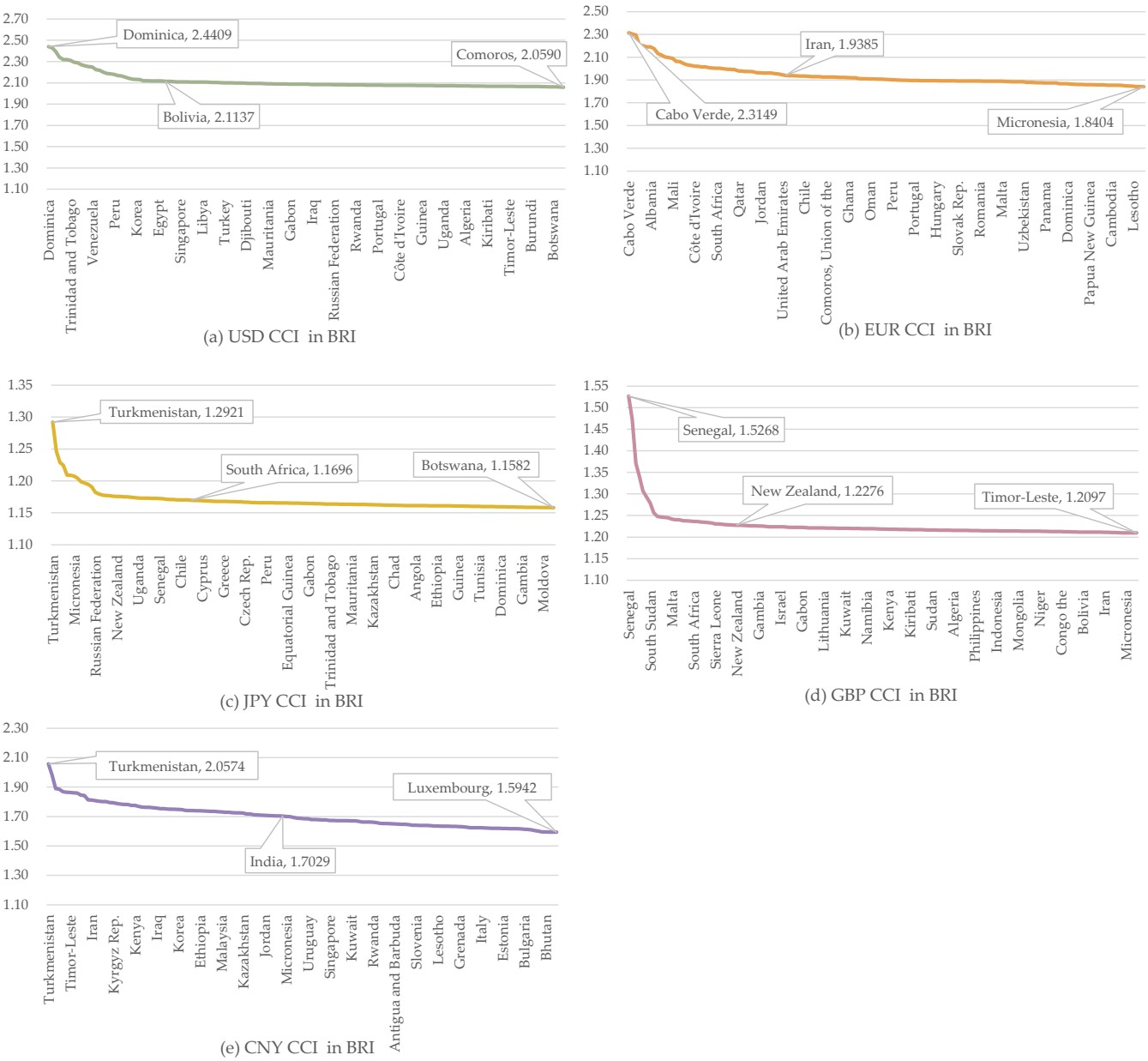

**Fig 2. CCI distribution in BRI countries, 2020Q4.** (a) USD CCI distribution, (b) EUR CCI distribution, (c) JPY CCI distribution, (d) GBP CCI distribution, (e) CNY CCI distribution. X-axis is arranged left to right according to countries with the highest to lowest CCI for each currency.

## 3.2 Forecasting results

We use ARIMA to expand the CCI results from 2006Q1 to 2020Q4 to the period 1996Q1–2020Q4 to forecast the trends of these currencies. ARIMA models are commonly used to forecast and model time-series data, allowing for the prediction of future trends and changes by analyzing historical data. We assess the stationarity of the five currencies using the unit-root test. Table 6 presents the identification and order determination, parameter estimation, and model testing for the ARIMA models associated with the five CCIs.

**Table 6. ARIMA models of five CCIs.**

|  | USD CCI | EUR CCI | JPY CCI | GBP CCI | CNY CCI |
|---|---|---|---|---|---|
| ARIMA Model (p, d, q) | (0,1,1) | (1,0,1) | (1,1,1) | (1,1,1) | (1,1,1) |
| AR(1) |  | 0.7750*** | 0.8554*** | 0.7940*** | 0.7005*** |
|  |  | (0.0789) | (0.1209) | (0.0803) | (0.0912) |
| MA(1) | −0.5798*** | 0.3183*** | −0.9409*** | −1.0000 | −1.0000 |
|  | (0.0825) | (0.1181) | (0.0807) | (283.0297) | 305.3998 |
| Constant |  | 1.9710*** |  | −0.0038*** | 0.0091*** |
|  |  | (0.0626) |  | (0.0008) | (0.0014) |
| Observations | 99 | 88 | 99 | 99 | 99 |
| AIC | −1.1761 | −1.6341 | −1.4316 | −2.9820 | −1.7410 |
| R-squared | 0.2611 | 0.7476 | 0.0231 | 0.1026 | 0.1454 |

Note: EUR data start from 1999, as the EUR was launched on January 1, 1999. Significance levels are denoted by ***, **, and *, representing 1%, 5%, and 10% significance, respectively, with the corresponding standard errors in parentheses.

We determine the suitable (p, q, d) parameters for the ARIMA model of each currency based on the outcomes of autocorrelation, partial autocorrelation, and AIC, as shown in the second row of Table 6. We can see that all currencies, except for EUR, demonstrate first-order stationarity. Furthermore, apart from USD, the other currencies incorporate an AR(1) term while all currencies feature an MA(1) term. The Q-test for residuals, which examines the autocorrelation coefficient of residuals being zero, upholds the null hypothesis for all models, signifying independent residuals and appropriate models for forecasting.

The predicted CCI rankings of the five currencies remain consistent with those of 2020Q4. USD retains its leading position in currency competitiveness, closely trailed by EUR. CNY continues its upward trajectory, securing the third position, while GBP shows a slight declining trend, positioning it at a level akin to that of JPY in terms of currency competitiveness (Fig 3).

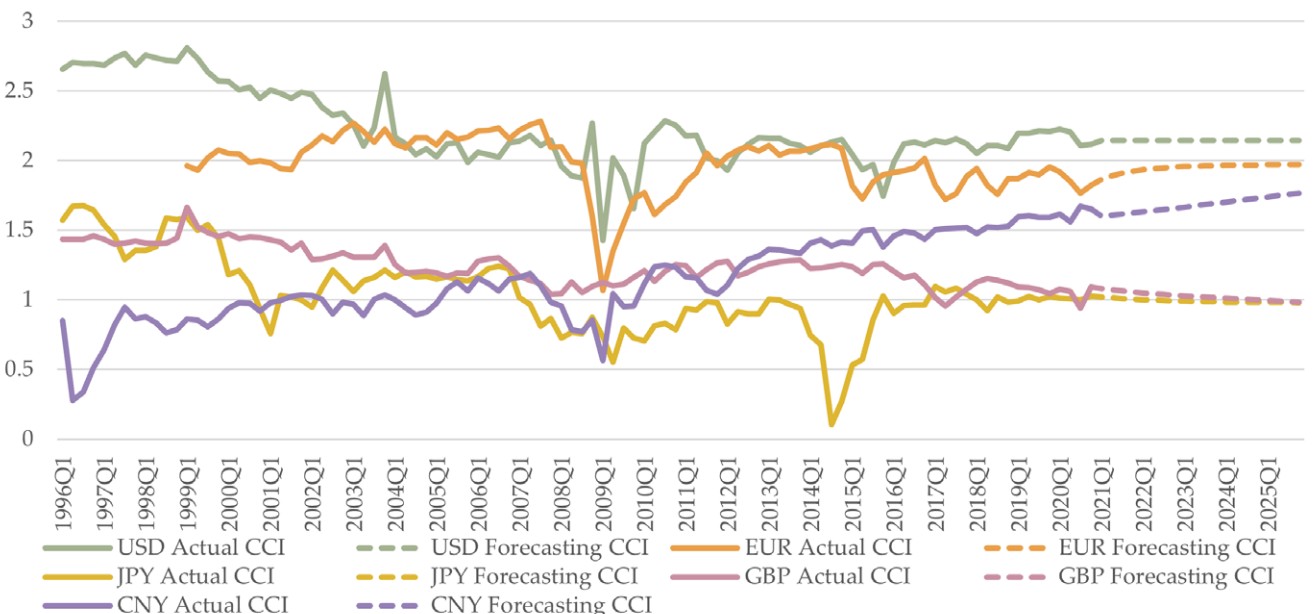

**Fig 3. Actual and forecasted values of five CCIs.** The solid line represents the actual values from 1996Q1 to 2020Q4; the dashed line represents the predicted values from 2021Q1 to 2025Q4.

## 4. Discussion

Among the five examined currencies, the USD emerges as the most competitive currency among BRI countries, closely followed by the EUR, which demonstrates greater volatility. The CNY is steadily increasing in competitiveness while the JPY exhibits lower influence compared with the USD and EUR. Meanwhile, the GBP, although stable, lags behind in competitiveness in BRI countries. Consequently, the USD and EUR are considered low-risk currency options, CNY shows potential for growth, and JPY and GBP are considered less attractive.

This study's main innovations are as follows: First, we adopt a currency perspective to address gaps in the literature, particularly concerning international currency selection for managing debtor risk. While currency selection significantly affects the management of currency risk for debtor countries, few studies have focused on this aspect. Second, we use CCI as a metric to gauge a currency's influence or performance in a specific region. Different from using the level of internationalization as a currency performance indicator, currency competitiveness provides a more accurate representation of currency influence. Third, we focus on BRI countries—an area often overlooked in currency competitiveness studies. By comparing the currency competitiveness of the USD, EUR, JPY, CNY, and GBP in 142 BRI countries, we address issues pertaining to currency selection and combination, offering suggestions for mitigating currency risk and fostering the sustainable development of BRI countries.

### 4.1 Time perspective on the five main currencies

**4.1.1 CCI before and after the BRI.** Comparing CCI before and after the BRI offers insights into changes in the competitiveness of these currencies (Fig 4). USD maintains consistently high competitiveness throughout the period, indicating that the BRI does not significantly disrupt USD competitiveness in BRI countries. EUR shows slightly stronger

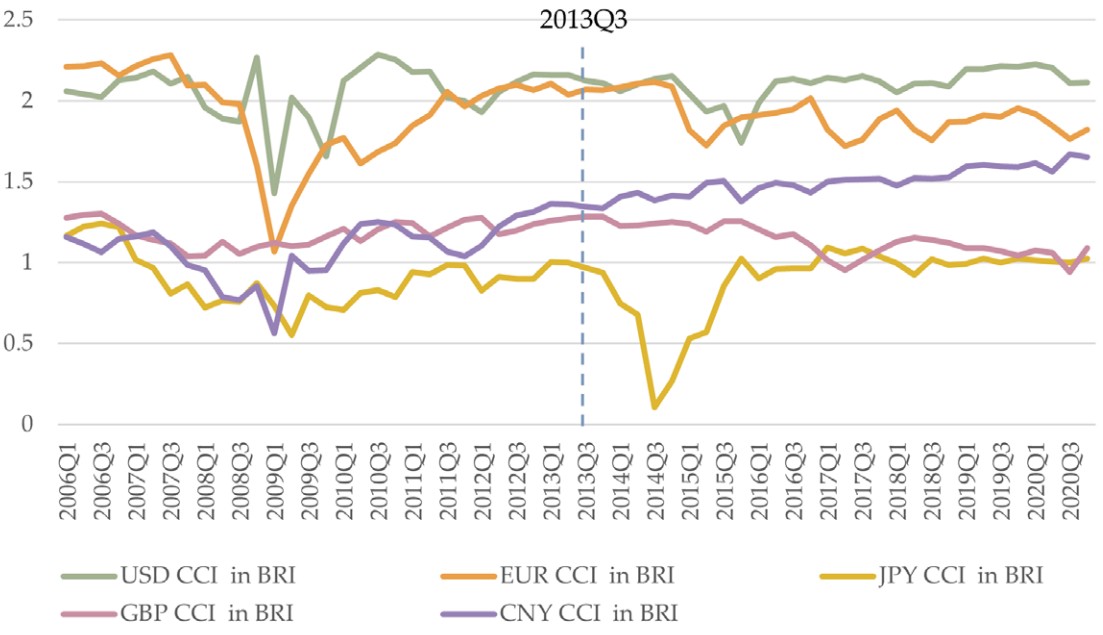

**Fig 4. CCI trends of USD, EUR, JPY, GBP, and CNY in the BRI area from 2006Q1 to 2020Q4, with a dotted reference line at 2013Q3 when BRI was proposed.** For comparison, we use equidistant time intervals for the periods before and after the BRI. The time span before the BRI (2006Q1 to 2013Q2, 30 quarters) is equivalent to the time span after the BRI (2013Q3 to 2020Q4, 30 quarters).

competitiveness before the BRI than after, with some fluctuations. JPY exhibits more stability after the BRI than before, with a general increase, except for a sudden drop in 2014. GBP demonstrates the most stable trend among the five currencies over the period, with a slight decline after the BRI compared with a slight increase before. CNY shows increasing volatility, moving from fifth to third place by 2020Q4.

The landscape of monetary competition has changed since the introduction of the BRI. The most noticeable change is the rapid and steady growth of CNY, with some limited strengthening of EUR and JPY. However, BRI introduction has had little effect on USD and GBP. This result aligns with studies suggesting that the BRI has strengthened CNY [64,65]. However, there are some differences. First, although CNY has been strengthened, it is not as exaggerated as some sources claim [66,67], and it is only in the third position, with a significant gap from the first two. Second, other studies do not compare the growth rate of CNY competitiveness before and after the BRI. Our study does, however, and finds that the BRI effectively promotes CNY. Third, while studies have mainly considered the BRI's effect on CNY, we find that the BRI also promotes EUR and JPY, contrary to previous studies [68,69]. Based on these findings, we conclude that USD remains strong, EUR is second but strengthened by the BRI, CNY has potential, JPY is stronger than GBP, and the degree of risk increases in that order.

**4.1.2 CCI at present and in the future.** Currency competitiveness reflects a country's overall strength, encompassing both past performance and future expectations. Currencies with high competitiveness and influence exhibit greater stability and market confidence, resulting in lower currency risk. Using a high CCI currency for international transactions can effectively mitigate currency risk for both parties. A currency maintaining a consistently high CCI is perceived as having lower risk in the future. Therefore, BRI countries can reduce their currency risk by selecting a currency with a high current and future CCI.

Considering current competitiveness levels and future trends, USD emerges as the most competitive currency in BRI countries, followed by EUR and then CNY. USD stands out as the most stable currency while EUR is a favorable alternative from a temporal perspective. China's CNY, a developing-country currency with a growing international presence, might become increasingly significant in global markets, making it a viable choice for BRI countries. The collaborative effort involved in building the BRI and China's enhanced communication with BRI countries contribute to the rising CCI of CNY. Therefore, CNY presents itself as a competitive, low-risk currency option for BRI countries. Meanwhile, GBP and JPY are not as competitive for BRI countries. Overall, USD will remain the strongest currency in the future, and it will be difficult for EUR to overtake it [70,71]. As for CNY, while it is still a long way from becoming an international currency [72,73], its competitiveness is growing, and there is a clear trend toward regionalization. As a result, USD will remain the least risky currency going forward, followed by EUR. JPY and GBP are stable but not as competitive. CNY has great potential, but there are some risks.

## 4.2 Spatial perspective on the five main currencies

From a spatial economics perspective, space encompasses both geographical and economic dimensions, with economic space becoming increasingly significant [42]. Economic spatial relationships play a crucial role in currency selection, as highlighted by international currency selection theory [74–76]. Opting for a currency from a country with strong economic and trade ties can help mitigate risk [77,78]. Central banks consider currencies with low risk and stability for reserves while also favoring currencies from countries with extensive bilateral transactions [79]. This approach is particularly beneficial for traders, investors, and debtors, as it minimizes currency risk and avoids mismatches.

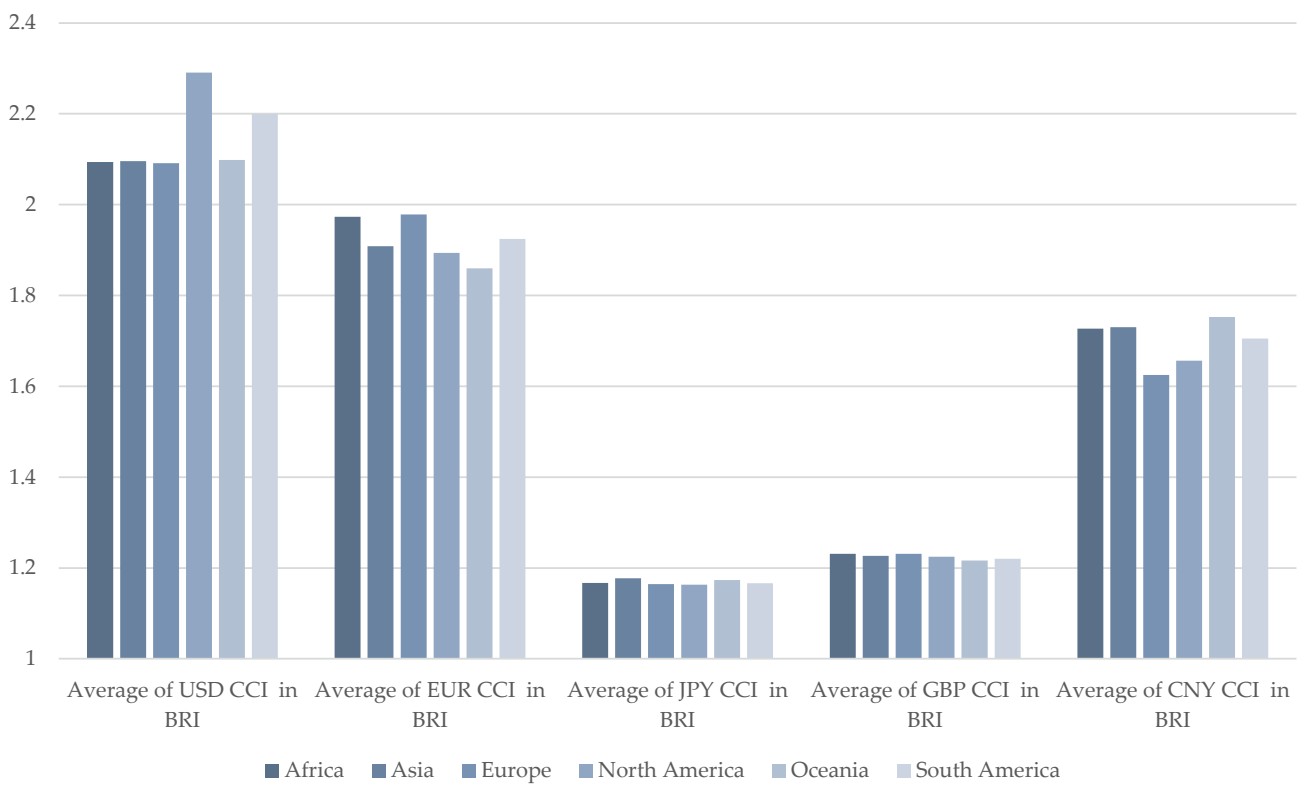

**Fig 5. CCI of continental groups by currency in 2020Q4.**

To better investigate currency selection for BRI countries, we group 142 BRI countries in two dimensions: by continent and by currency competition status. We also examine BRI countries' currency choices for risk prevention through the lens of economic spatial relations. This analysis focuses on bilateral trade and investment relations, based on discussions of bilateral relations and currency selection, and uses trade data from the IMF DOTS database to gauge trade relations' proximity. We use direct investment position data from the IMF CDIS database to assess investment relationships.

**4.2.1 Analysis of BRI countries by continental group.** The 142 BRI countries are organized into six groups by continent, the same as in section 3.1.2. Our comparative analysis focuses on currency competitiveness at the continental level and the continent's bilateral relations with the five currency-issuing countries. When comparing continent-level currency competitiveness, CCI in 2020Q4, averaged across continents, reveals that the competitiveness of each currency is strong in different geographic areas. USD is the most competitive overall and is better in North and South America. EUR is second to the USD in overall competitiveness and is more competitive in Africa and Europe while CNY is third in overall competitiveness and is less competitive in Europe and North America. GBP and JPY are generally less competitive, with smaller differences between continents (Fig 5). This is consistent with what is described in the literature—namely, that geographical distance influences currency competitiveness. For currencies whose influence breaks through political territories, the closer the geographical proximity, the more competitive they are [80]. However, CNY seems to be an exception, as its high CCI in Africa does not align with what geographical distance implies. African countries are far away from China, but CNY is competitive in this continent. This

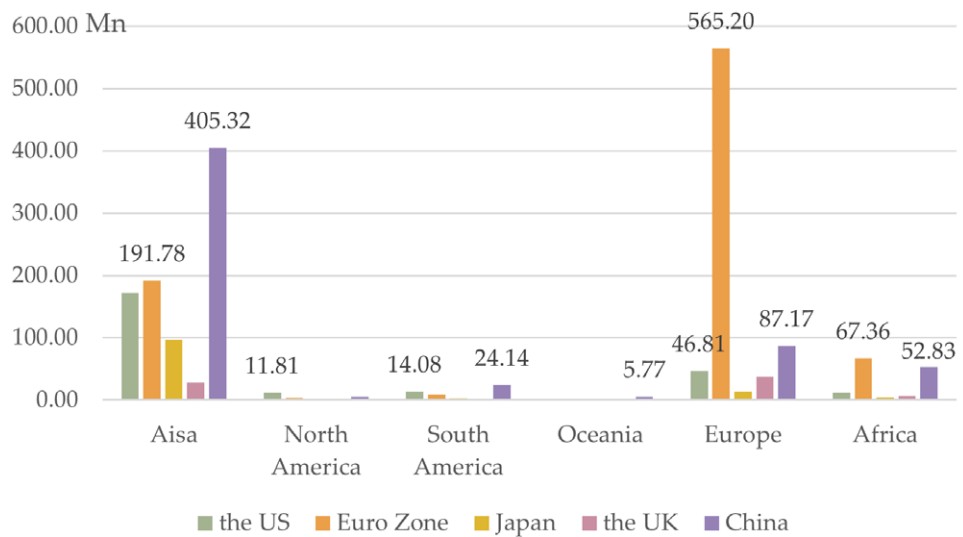

**Fig 6. Bilateral trade distribution by continent in BRI countries in 2020Q4.** Pricing unit: Million USD.

could be attributable to the BRI's effects on bilateral relations, as noted in the literature [81–83].

We categorize the 142 BRI countries by continent to analyze bilateral trade relations from a continental perspective. Fig 6 shows the bilateral trade relationships. In 2020Q4, in BRI Asia, the closest bilateral trade relations are with China, with a trading volume of 405.32 million USD, surpassing the combined trade volumes with the Eurozone (191.78 million USD) and the US (172.12 million USD). North American BRI countries predominantly trade with the US, with a trading volume that is 11.81 million USD higher than that with China (5.37 million USD) and the Eurozone (3.63 million USD), making USD the preferred choice for these countries, followed by CNY and EUR. Trade volume between South American BRI countries and China reached 24.14 million USD, surpassing trade volumes with the US (14.08 million USD) and the Eurozone (8.98 million USD). Bilateral trade volumes between BRI Oceania countries and each currency issuer are relatively low, except for China, with a trade volume of 5.37 million USD. European BRI countries exhibit the highest bilateral trade volume with the Eurozone, followed by China and the US. In African BRI countries, trade volume with the Eurozone slightly exceeds that with China and significantly surpasses that with the US, Japan, and the UK.

Fig 7 illustrates the bilateral investment relationships among BRI continental groups, with the Eurozone emerging as the preferred choice owing to its closer investment relationship compared with other currency issuers, excluding Oceania countries close to China. Notably, Asian, North American, South American, and European BRI countries exhibit strong investment ties with the US while African nations have closer investment relationships with China, consistent with the literature [84].

We find that bilateral relations are characterized by a near-neighborhood effect and BRI effect. For the near-neighborhood effect, Asian countries and China, European countries and the Eurozone, and African countries and the Eurozone have close bilateral trade relations. This reflects the regional economics principle that trade becomes more convenient and bilateral relations become closer with reduced distance between countries [85,86]. For the BRI effect, South American and African countries have close relations with China, which does not

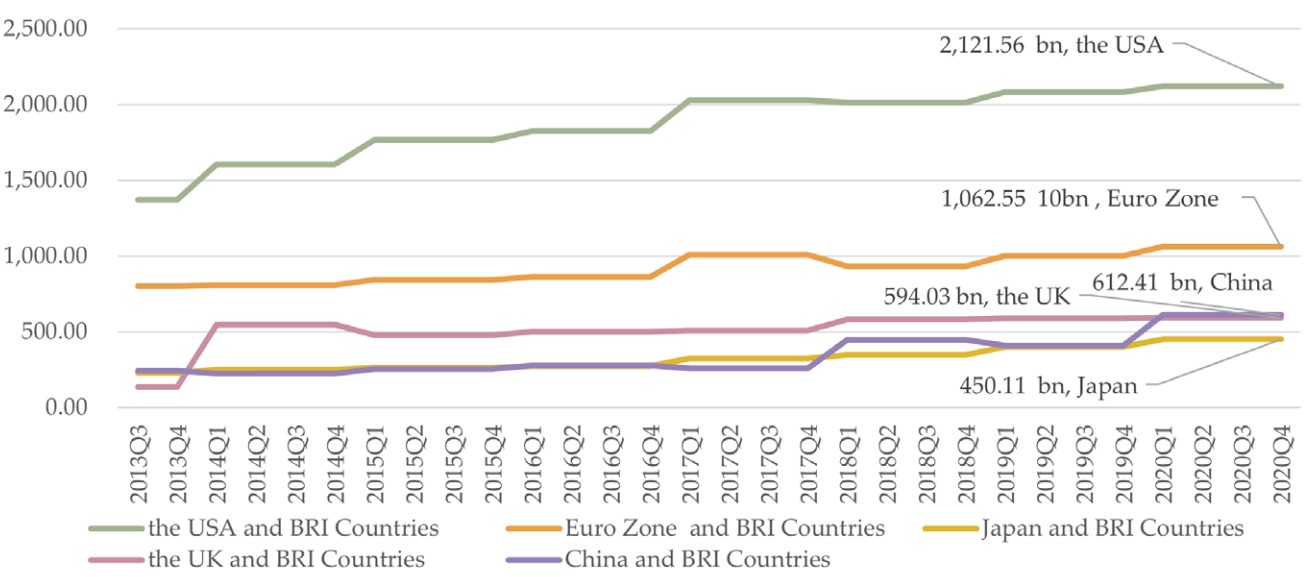

**Fig 7. Bilateral direct investment position distribution by continent among BRI countries in 2020Q4.** Pricing unit: Billion USD. Data source: IMF CDIS database.

follow the rule regarding distance but reflects the role of the BRI in promoting bilateral development. Hence, the BRI effect helps explain exceptions to the regional economics principle.

Third, we examine currency selection in the context of CCI continental features and bilateral relations. Countries in North America near the US exhibit close bilateral relations with the US, suggesting USD as the optimal currency choice. Similarly, countries in Europe and Africa near the Eurozone have stronger trade and investment ties with the Eurozone, making EUR the preferred currency selection. Yet, as the CCI and bilateral relations of African countries and China increase, CNY becomes a suboptimal option. Asian countries near China and countries in South America and Oceania are under the BRI effect and show significant bilateral trade volumes with China, indicating CNY as a suitable currency option for these regions. However, for Asian countries, since they have deep investment relationships with the Eurozone, EUR is another optimal option. In this context, overall, EUR stands as the top choice for most BRI countries to mitigate currency risk, followed by CNY and USD. Conversely, JPY and GBP are less favorable options for BRI countries.

**4.2.2 Analysis of BRI countries by currency competition status groups.** First, we group BRI countries according to their currency competition status, which is obtained by categorizing the results based on the labeling method. This method uses the mean and extreme variance as a guide, with 10% above the mean and 0.5% below the mean as the criteria for segmentation. The portion above 10% of the mean is labeled "very competitive," that within 10% is "competitive," that within −0.5% is "less competitive," and that below −0.5% is "uncompetitive." After labeling, each country has five tags. We obtain three categories of currency competition status: countries with balanced currencies (CBCs), countries with several major currencies (CSMCs), and countries with one super currency (COSCs). CBCs refer to countries without currencies at the competitive level, indicating a lack of fierce currency competition. Currency competition in CMSCs is intense, with at least two currencies at the competitive level or above. In COSCs, currency competition is not fierce since there is only one very competitive or competitive currency in the country. Table 7 shows that 48.59% of all BRI countries are CSMCs, 39.44% are

**Table 7. BRI countries grouped by currency competition status.**

| Country Group | Continent | Countries | Total |
|---|---|---|---|
| Countries with balanced currencies (CBCs) | Africa | Zambia, Rwanda, Namibia, Mozambique, Lesotho, Burundi, Botswana | 7 |
| | Asia | Afghanistan, Nepal, Georgia, Bhutan | 4 |
| | Europe | Romania, Poland, Czech Republic, Bulgaria | 5 |
| | South America | Bolivia | 1 |
| Countries with several major currencies (CSMCs) | Africa | Egypt, Angola, South Sudan, Somalia, South Africa, Nigeria, Niger, Libya, Equatorial Guinea, Chad, Zimbabwe, Sierra Leone, Seychelles, Senegal, Mauritania, Madagascar, Liberia, Kenya, Guinea, Gabon, Ethiopia, Djibouti, Congo, Cameroon | 24 |
| | Asia | Turkmenistan, Saudi Arabia, Israel, Kuwait, Yemen, Korea, India, United Arab Emirates, Qatar, Philippines, Pakistan, Malaysia, Indonesia, Vietnam, Turkey, Thailand, Mongolia, Jordan, Myanmar, Brunei Darussalam, Bangladesh, Bahrain | 22 |
| | Europe | Ukraine, Russian Federation, Estonia, Belarus, Malta, Greece, Cyprus | 7 |
| | North America | Trinidad and Tobago, Panama, Grenada, Barbados, Antigua and Barbuda | 5 |
| | Oceania | Micronesia, New Zealand, Tonga, Papua New Guinea, Kiribati | 5 |
| | South America | Suriname, Chile, Venezuela, Peru, Guyana, Ecuador | 6 |
| Countries with one super currency (COSCs) | Africa | Uganda, Tunisia, Togo, Tanzania, Sudan, Morocco, Mali, Ghana, Gambia, Côte d'Ivoire, Congo, Comoros, Cabo Verde, Benin, Algeria | 15 |
| | Asia | Singapore, Uzbekistan, Timor-Leste, Tajikistan, Sri Lanka, Oman, Maldives, Lebanon, Lao, Kyrgyzstan, Kazakhstan, Iraq, Iran, Cambodia, Azerbaijan, Armenia | 16 |
| | Europe | Slovakia, Slovenia, Portugal, Luxembourg, Lithuania, Latvia, Italy, Austria, Serbia, Montenegro, Moldova, North Macedonia, Hungary, Bosnia and Herzegovina, Albania, | 15 |
| | North America | Jamaica, El Salvador, Costa Rica, Dominican Rep., Dominica | 5 |
| | Oceania | Vanuatu, Solomon Islands, Samoa, Fiji | 4 |
| | South America | Uruguay | 1 |

CBCs, and 11.97% are COSCs. Thus, we can conclude that currency competition is fierce in most BRI countries.

For CBCs and COSCs, their currency selection is clear. CBCs do not have competitive currencies. Their currency selection could be based on their continent, as discussed in section 4.2.1. COSCs have one super currency, which would be the optimal currency.

It is necessary to take a close look at bilateral relations among CSMCs. Fig 8 shows the bilateral trade relations of CMSCs with the five currency-issuing countries. We can see that CSMCs have the strongest bilateral trade relations with China, especially Asian CSMCs. Next, CSMCs have the strongest bilateral trade relations with the Eurozone, followed by the US and Japan, with the UK last. Fig 9 illustrates the bilateral investment relationships of CMSCs with the five currency-issuing countries. We can see that CSMCs have the strongest relationship with the Eurozone, followed by the US, and slightly stronger relationships with China than with Japan and the UK. It seems clear, then, that for trade-oriented countries, CNY is the best option, while for investment-oriented countries, EUR is the best option. This can be understood as CNY being better for developing countries and EUR being better for developed countries. This finding builds on Claessens's argument that the ideal foreign currency composition for a developing nation is contingent on the correlation between export revenues and borrowing costs in each currency, along with the relationship between borrowing costs across different currencies [27]. We argue that when there are multiple competing currencies, CNY is the optimal choice for developing BRI countries because of China's close trade relations with them

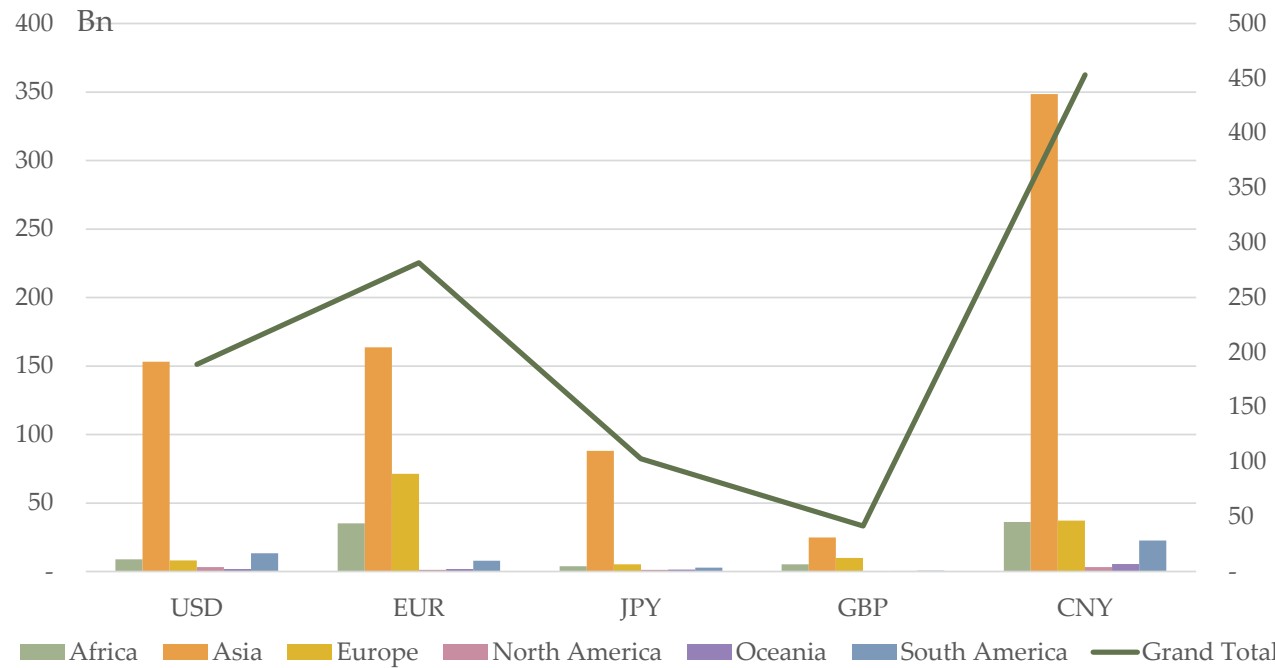

**Fig 8. Bilateral trade distribution of CSMCs in 2020Q4.**

and because of the reduced risk and lower borrowing costs as CNY's competitiveness increases. Since developed BRI countries are mainly in Europe, they have close trade relations with EUR, and EUR has low local borrowing costs; thus, EUR is the optimal choice for developed BRI countries. USD, as a vehicle currency, is the alternative for BRI countries.

## 5. Conclusion

This study explores currency risk in BRI countries from a currency competition perspective, assessing the competitiveness of the USD, EUR, JPY, GBP, and CNY based on CCI and analyzing spatiotemporal trends in the CCI results. Temporally, we contrast CCI findings before and after BRI introduction and consider current and future scenarios, noting slight fluctuations but overall stability for USD and GBP. Significantly, CNY, EUR, and JPY show improved competitiveness under the BRI, with CNY showing the most significant enhancement and maintaining an upward trend, while EUR and JPY receive comparatively limited boosts. Spatially, we categorize 142 BRI countries based on geographical location and currency competitiveness, examining bilateral relationships between the currency-issuing countries and the grouped nations. It is recommended that North American countries opt for either USD or EUR, European countries opt for EUR, and nations in Africa, Asia, South America, and Oceania opt for EUR and CNY, with USD as a secondary choice. Highly competitive currencies are recommended for certain countries while others should make choices based on continental affiliations or development status.

USD and EUR emerge as stable and competitive currency options for BRI nations while the rising competitiveness of CNY and strengthened ties between China and BRI countries could position CNY as a top contender. Although GBP and JPY remain stable but less competitive, they could still be viable choices depending on specific country circumstances. These findings

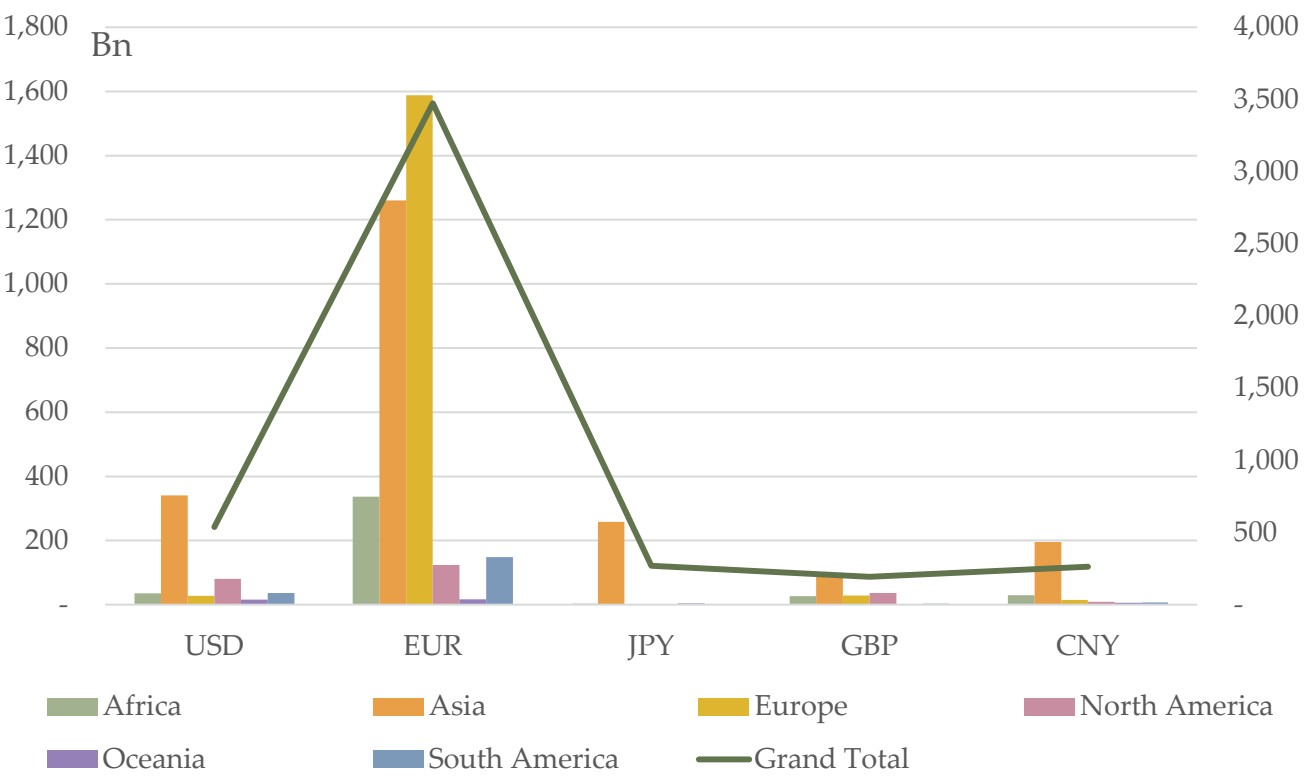

**Fig 9. Bilateral investment distribution of CSMCs in 2020Q4.**

can help BRI countries mitigate currency risks through strategic currency selection, thus enhancing debt sustainability.

Policy recommendations include prioritizing geographic location and bilateral relations with currency-issuing countries for countries lacking competitive currencies, with USD as a secondary option. Dominant-currency countries are advised to diversify risks by choosing currencies from nearby regions with robust bilateral relationships. Additionally, CNY could be the optimal choice for developing BRI nations with multiple competing currencies while EUR is recommended for developed counterparts. Strengthening financial markets supports foreign exchange management, contributing to sustainable debt management.

This study contributes to the literature on debt sustainability by integrating currency risk and choice perspectives, extending the scope to 142 BRI countries. It is necessary, however, to further consider aspects such as the effect of each BRI country's population on currency choices. The effect of the COVID-19 pandemic on currency selection should also be considered. Finally, influential currencies such as the Canadian dollar, Russian ruble, and Swiss franc could be included in future research.

## Supporting information

**S1 Data. Indicators of Currency Competitiveness Index (CCI), bilateral trade volume, and bilateral direct investment position.**
(XLSX)

## Author Contributions

**Conceptualization:** Mengyang Li, Lei Wang.

**Data curation:** Mengyang Li.

**Formal analysis:** Mengyang Li.

**Investigation:** Mengyang Li.

**Methodology:** Mengyang Li, Wencui Liu.

**Project administration:** Wencui Liu.

**Resources:** Mengyang Li.

**Software:** Mengyang Li.

**Supervision:** Lei Wang.

**Validation:** Mengyang Li, Lei Wang.

**Visualization:** Mengyang Li.

**Writing – original draft:** Mengyang Li.

**Writing – review & editing:** Mengyang Li, Lei Wang.

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
