## [Decision Letter · Decision Letter 0]

8 Apr 2024

PONE-D-24-08628A Currency Risk Analysis of Belt and Road Initiative CountriesPLOS ONE

Dear Dr. Wang,

Thank you for submitting your manuscript to PLOS ONE. After careful consideration, we feel that it has merit but does not fully meet PLOS ONE’s publication criteria as it currently stands. Therefore, we invite you to submit a revised version of the manuscript that addresses the points raised during the review process.

**ACADEMIC EDITOR:  ** Both reviewers have recommended the same: major revisions. Please carefully review all comments made by the two reviewers and revise accordingly. Additionally, the revised manuscript should be edited and proofread by professional English editing services to ensure the final text is flawless in terms of writing quality.

We look forward to receiving your revised manuscript.

Kind regards,

Poowin Bunyavejchewin

Academic Editor

PLOS ONE

“the State Key Laboratory of Desert and Oasis Ecology: Xinjiang Institute of Ecology and Geography: CAS (E1510107)”

3. We note that Figures 3,7,8 and 10 in your submission contain [map/satellite] images which may be copyrighted. All PLOS content is published under the Creative Commons Attribution License (CC BY 4.0), which means that the manuscript, images, and Supporting Information files will be freely available online, and any third party is permitted to access, download, copy, distribute, and use these materials in any way, even commercially, with proper attribution. For these reasons, we cannot publish previously copyrighted maps or satellite images created using proprietary data, such as Google software (Google Maps, Street View, and Earth). For more information, see our copyright guidelines: http://journals.plos.org/plosone/s/licenses-and-copyright.

a. You may seek permission from the original copyright holder of Figures 3,7,8 and 10 to publish the content specifically under the CC BY 4.0 license. 

4. We are unable to open your Supporting Information file [PlosoneData.7z]. Please kindly revise as necessary and re-upload.

Reviewers' comments:

Reviewer's Responses to Questions

**Comments to the Author**

1. Is the manuscript technically sound, and do the data support the conclusions?

Reviewer #1: Partly

Reviewer #2: Yes

2. Has the statistical analysis been performed appropriately and rigorously? 

Reviewer #1: Yes

Reviewer #2: I Don't Know

3. Have the authors made all data underlying the findings in their manuscript fully available?

Reviewer #1: Yes

Reviewer #2: Yes

4. Is the manuscript presented in an intelligible fashion and written in standard English?

Reviewer #1: Yes

Reviewer #2: Yes

5. Review Comments to the Author

Reviewer #1: I found this study very interesting as it provides valuable insights into currency risk analysis in Belt and Road Initiative countries. Nonetheless, I suggest authors consider the following comments to significantly improve the overall quality of their manuscript.

1/ Authors should add a section on the literature review establishing the theoretical context and describing the results of previous studies on the subject.

2/ You must provide references or justification supporting each of the thirteen variables involved in the construction of the CCI index (see table 1) and each equation/model.

3/When discussing the results, add additional references and stylized facts that explain/justify the results. You should also compare your results to those of previous studies.

4/The conclusion should also present the detailed policy and practical implications of your findings and highlight the limitations of the study and avenues of research for future analyses.

Reviewer #2: The relevance of the topic is well presented. It deals with an interesting, timely and economically beneficial topic, but it needs significant development. My specific suggestions are:

The literature review is very superficial, only tangentially referring to the debates and methodologies, not presenting the research niche in depth.

What is missing from the literature review is a more detailed description of the other methodologies that have been used to research the topic. From this, it could be deduced why the author uses this methodology to answer his research question. I miss the operationalisation derived from the theoretical framework.

I would suggest a more in-depth presentation of the theoretical background that underpins this methodology.

It refers to different models (e.g. Anchor Effect of Currency), but we do not know in depth what those models say. In the theoretical introduction, the models used and their main findings should be discussed, so that we can later better understand the results that are compared with them.

From row 92, the description of the indicators should also be provided in Table 1. There is no detailed description that would justify a textual description.

The content of Table 2 (data sources) should be included in Table 1, two tables are unnecessary.

Can you provide data on the original database, the number of unestimated observations per variable? Give us a better insight into the database.

Improve the graphical appearance of the charts, they are now very undemanding.

Figures 3, 7, and 10 are not suitable for comparison in their present form. This can be submitted as an appendix, but I would suggest that the authors cluster the countries based on the results. It would be much better to illustrate their results. In the text, they could explain the reasons behind the clusters.

In line 303, the author refers to "discussions on currency competitiveness", but we do not find anything about them in the theoretical background.

The conclusion is very brief. More detail should be given on the results of this research, what it adds to the literature, what its limitations are, and what other new research directions the authors have discovered during their research. What region-specific policy recommendations can they make based on their findings.

For Editors: The methodology seems right to me, but I'm not familiar with it, so it's worth asking other reviewers. I'm not a native English speaker, so I can't fully judge the correctness of the language. It was understandable to me.

6. PLOS authors have the option to publish the peer review history of their article (what does this mean?). If published, this will include your full peer review and any attached files.

Reviewer #1: No

Reviewer #2: No

---

## [Author Response · Author response to Decision Letter 0]

18 May 2024

Response to Editor’s comments

Comment 1. Please ensure that your manuscript meets PLOS ONE's style requirements, including those for file naming. The PLOS ONE style templates can be found at

Response: We have carefully reviewed PLOS ONE's style requirements, including those for file naming, and have made the necessary adjustments. Our revised manu-script now adheres to all the specified guidelines.

Comment 2. Thank you for stating the following financial disclosure:

“the State Key Laboratory of Desert and Oasis Ecology: Xinjiang Institute of Ecology and Geography, CAS (E1510107)”

Response: Regarding the financial disclosure provided, we acknowledge the support of the State Key Laboratory of Desert and Oasis Ecology: Xinjiang Institute of Ecol-ogy and Geography, CAS. In accordance with your request, we confirm that the fun-ders played no role in the study design, data collection and analysis, decision to pub-lish, or preparation of the manuscript.

Comment 3. We note that Figures 3,7,8 and 10 in your submission contain [map/satellite] images which may be copyrighted. All PLOS content is published under the Creative Commons Attribution License (CC BY 4.0), which means that the manu-script, images, and Supporting Information files will be freely available online, and any third party is permitted to access, download, copy, distribute, and use these materials in any way, even commercially, with proper attribution. For these reasons, we cannot pub-lish previously copyrighted maps or satellite images created using proprietary data, such as Google software (Google Maps, Street View, and Earth). For more information, see our copyright guidelines: http://journals.plos.org/plosone/s/licenses-and-copyright.

We require you to either (1) present written permission from the copyright holder to publish these figures specifically under the CC BY 4.0 license, or (2) remove the figures from your submission.

Response: We have addressed the issue raised by the editor and reviewer by replacing Figures 3, 7, 8, and 10 with new figures. The original figures were based on standard map production with the map approval number GS(2016)1667 from the Chinese Min-istry of Natural Resources (http://bzdt.ch.mnr.gov.cn/), and the basemap remains un-altered. We have also improved our result displays and discussions as suggested by Reviewer #2, who pointed out the shortcomings of the original figures in making a comparison between five currencies. In response to this feedback, we have tried an-other way to display our results and supporting data for our discussion. We have grouped the results from 142 countries by a new method that labels them according to their currency competition status and further discuss these groups, thereby improving our paper.

Comment 4. We are unable to open your Supporting Information file [PlosoneData.7z]. Please kindly revise as necessary and re-upload.

Response: We have re-uploaded our data file. We hope that it can be opened success-fully this time.

Response to reviewer’s comments

Reviewer #1

Comment 1. I found this study very interesting as it provides valuable insights into currency risk analysis in Belt and Road Initiative countries. Nonetheless, I suggest authors consider the following comments to significantly improve the overall quality of their manuscript.

Response: Thank you for your valuable feedback. We appreciate your positive com-ments on the study and are grateful for your suggestions to improve the overall quality of our manuscript. We will carefully considered and addressed the comments you have provided to enhance the value and impact of our research.

Comment 2. Authors should add a section on the literature review establishing the theoretical context and describing the results of previous studies on the subject.

Response: Thank you for your feedback. We have addressed your suggestion by in-cluding a section on the literature review to establish the theoretical context and de-scribe the results of previous studies on the subject. Specifically, we have added three sections on the literature review, covering debt sustainability issues, currency risk, and currency competition.

The literature review on debt sustainability issues explores the existing research on debtors’ solvency and economic stability, while also highlighting the significance of currency risk. This can be found in lines 51-72 of the manuscript.

The literature review on currency risk is organized into two dimensions and can be found in lines 73-106.

Additionally, we have included a section on the currency competition theory as a new entry point for our research, which can be found in lines 107-140.

We believe that these additions provide a comprehensive theoretical context and an overview of previous studies on the subject. Thank you for your valuable input.

Comment 3. You must provide references or justification supporting each of the thir-teen variables involved in the construction of the CCI index (see table 1) and each equation/model. 

Response: Included references for each variable used in our CCI index, and the mod-ifications made can be found in lines 161-183 and 212-216.

Comment 4. When discussing the results, add additional references and stylized facts that explain/justify the results. You should also compare your results to those of pre-vious studies. 

Response: Additional references and stylized facts have been incorporated into the Discussion section to further explain and justify the results. The characteristics of cur-rency competitiveness, bilateral trade, and bilateral investment are summarized using the data, and the results are compared with previous studies.

We have added facts about the characteristics of CCI before and after BRI and com-pared them with the literature (Lines 443-456).

The characteristics of CCI at present and in future expectations have been included, aligning with the literature (Lines 474-477).

A comparison of the results of this paper with the literature in the continental group-ings has been added (Lines 544-549).

We have included a comparison between the outcomes of this study and those in the existing literature, demonstrating similar trends but with more detailed findings (Lines 599-609).

Comment 5. The conclusion should also present the detailed policy and practical im-plications of your findings and highlight the limitations of the study and avenues of research for future analyses.

Response: The conclusions have been revised to address the specific concerns raised by the reviewer. We have provided more detailed policy and practical implications of our findings, as well as highlighted the limitations of the study and avenues for future research. Specifically, we have revised the presentation of results in the conclusions to provide more detail on the research results (Lines 613-633), added policy recommen-dations and discussed the marginal contribution of the paper (Lines 634-641), and in-cluded content on limitations and future research directions (Lines 642-647). Thank you for the valuable feedback, and we believe these revisions have strengthened the conclusions of the paper.

Reviewer #2

Comment 1. The relevance of the topic is well presented. It deals with an interesting, timely and economically beneficial topic, but it needs significant development. 

Response: Thank you for your feedback. I appreciate your recognition of the rele-vance of the topic. I I have provided a more comprehensive and in-depth analysis.

Comment 2. The literature review is very superficial, only tangentially referring to the debates and methodologies, not presenting the research niche in depth. What is miss-ing from the literature review is a more detailed description of the other methodolo-gies that have been used to research the topic. From this, it could be deduced why the author uses this methodology to answer his research question. I miss the operationali-sation derived from the theoretical framework. I would suggest a more in-depth presentation of the theoretical background that underpins this methodology. It refers to different models (e.g. Anchor Effect of Currency), but we do not know in depth what those models say. In the theoretical introduction, the models used and their main findings should be discussed, so that we can later better understand the results that are compared with them.

Response: Thank you for your suggestion. We have supplemented the literature re-view in the introduction with three sections, debt sustainability issues, currency risk, and currency competition. Lines 51-72 provide a compendium of literature perspec-tives on debt sustainability issues, lines 73-106 introduce the content of the literature studies on currency risk, and lines 107-140 review the literature on currency competi-tion used in this paper.

In addition, we have added a brief introduction to the Anchor Effect of Currency to the literature review in lines 125-127, which is described in detail in the Methods sec-tion in lines 178-211. We believe that these additions address the concerns raised and provide a more in-depth presentation of the theoretical background that underpins the methodology used in this research.

Comment 3. From row 92, the description of the indicators should also be provided in Table 1. There is no detailed description that would justify a textual description. The content of Table 2 (data sources) should be included in Table 1, two tables are unnecessary. Can you provide data on the original database, the number of unesti-mated observations per variable? Give us a better insight into the database. 

Response: Revised Tables 1 and 2 to address the reviewer’ s comments. The descrip-tion of the indicators has been added to Table 1, and the content of Table 2 (data sources) has been included in Table 1 as well. This eliminates the need for two sepa-rate tables. We have also provided citations for the 13 indicators in the textual de-scriptions.

Additionally, we have updated Table 2 to include information on the original database, as well as the number of unestimated observations per variable. This provides a better insight into the database.

Comment 4. Improve the graphical appearance of the charts, they are now very un-demanding. Figures 3, 7, and 10 are not suitable for comparison in their present form. This can be submitted as an appendix, but I would suggest that the authors cluster the countries based on the results. It would be much better to illustrate their results. In the text, they could explain the reasons behind the clusters.

Response: We have improved the graphical appearance of the charts by removing the map displays for Figures 3, 7, and 10, and made changes to the results display. We have presented the results in both temporal and spatial dimensions. Under the spatial dimension, we have modified the results display by showing the Top 20 CCI countries by currency and continent of the 4th quarter of 2020 (Table 5), and the CCI distribu-tion in BRI countries in the 4th quarter of 2020 ranked from highest to lowest for each currency (Fig 2), paving the way for the grouping by currency competition status in section 4.2.2.

In the Discussion, we have grouped the results of the study CCI in a spatial perspec-tive, retaining the original continent grouping and adding a new grouping on the state of currency competition. We have also added new discussion based on this grouping. We believe these changes have significantly improved the graphical appearance of the charts and the overall presentation of the results.

Comment 5. In line 303, the author refers to "discussions on currency competitive-ness", but we do not find anything about them in the theoretical background.

Response: Added a section on currency competition and competitiveness in the liter-ature review of the introduction, specifically in lines 107-140. Thank you for bringing this to our attention.

Comment 6. The conclusion is very brief. More detail should be given on the results of this research, what it adds to the literature, what its limitations are, and what other new research directions the authors have discovered during their research. What re-gion-specific policy recommendations can they make based on their findings.

Response: We have revised the conclusion to provide more detail on the research re-sults, specifically in lines 613-633. We have included region-specific policy recom-mendations and highlighted the paper’s contribution based on the findings in lines 634-641. Additionally, we have added content on the limitations of the study and future research directions in lines 642-647.

---

## [Decision Letter · Decision Letter 1]

25 Jun 2024

PONE-D-24-08628R1Currency risk analysis of belt and road initiative countriesPLOS ONE

Dear Dr. Wang,

Thank you for submitting your manuscript to PLOS ONE. After careful consideration, we feel that it has merit but does not fully meet PLOS ONE’s publication criteria as it currently stands. Therefore, we invite you to submit a revised version of the manuscript that addresses the points raised during the review process.

**ACADEMIC EDITOR:  **The quality of the English needs to be improved. Therefore, I request that you consult professional English-editing services (e.g., Editage, Enago, etc.) and obtain an **English-editing certificate** from the service provider to be submitted along with the edited version of this manuscript.==============================

We look forward to receiving your revised manuscript.

Kind regards,

Poowin Bunyavejchewin

Academic Editor

PLOS ONE

Journal Requirements:

Reviewers' comments:

Reviewer's Responses to Questions

**Comments to the Author**

1. If the authors have adequately addressed your comments raised in a previous round of review and you feel that this manuscript is now acceptable for publication, you may indicate that here to bypass the “Comments to the Author” section, enter your conflict of interest statement in the “Confidential to Editor” section, and submit your "Accept" recommendation.

Reviewer #1: All comments have been addressed

Reviewer #2: All comments have been addressed

2. Is the manuscript technically sound, and do the data support the conclusions?

Reviewer #1: Yes

Reviewer #2: (No Response)

3. Has the statistical analysis been performed appropriately and rigorously? 

Reviewer #1: Yes

Reviewer #2: (No Response)

4. Have the authors made all data underlying the findings in their manuscript fully available?

Reviewer #1: Yes

Reviewer #2: (No Response)

5. Is the manuscript presented in an intelligible fashion and written in standard English?

Reviewer #1: Yes

Reviewer #2: (No Response)

6. Review Comments to the Author

Reviewer #1: Thank you for considering all of my previous comments. The paper has substantially been improved. I have no further suggestions.

Reviewer #2: (No Response)

7. PLOS authors have the option to publish the peer review history of their article (what does this mean?). If published, this will include your full peer review and any attached files.

Reviewer #1: **Yes: **Diby Francois Kassi

Reviewer #2: No

---

## [Author Response · Author response to Decision Letter 1]

27 Jun 2024

Response to Editor’s comments

Comment 1.

ACADEMIC EDITOR: 

The quality of the English needs to be improved. Therefore, I request that you consult professional English-editing services (e.g., Editage, Enago, etc.) and obtain an Eng-lish-editing certificate from the service provider to be submitted along with the edited version of this manuscript.

Response: Thank you for your feedback. We have already engaged professional Eng-lish-editing service (LetPub) to improve the quality of the manuscript. We have ob-tained an English-editing certificate from the service provider and will submit it along with the edited version of the manuscript.

Response to reviewer’s comments

Reviewer #1 and #2

Comment 1. All comments have been addressed.

Response: Thank you for your feedback. We have carefully addressed all comments and made the necessary revisions. We appreciate your time and effort in reviewing our work.

---

## [Editor Report · Decision Letter 2]

1 Jul 2024

Currency risk analysis of belt and road initiative countries

PONE-D-24-08628R2

Dear Dr. Wang,

We’re pleased to inform you that your manuscript has been judged scientifically suitable for publication and will be formally accepted for publication once it meets all outstanding technical requirements.

Kind regards,

Poowin Bunyavejchewin

Academic Editor

PLOS ONE

Additional Editor Comments (optional):

Although I do not believe the English quality is flawless, the authors do possess an English-editing certificate. Therefore, it should be sufficient to recommend that the Editor-in-Chief accept this version of the manuscript.
---

## [Editor Report · Acceptance letter]

4 Jul 2024

PONE-D-24-08628R2 

PLOS ONE

Dear Dr. Wang, 

I'm pleased to inform you that your manuscript has been deemed suitable for publication in PLOS ONE. Congratulations! Your manuscript is now being handed over to our production team.

Kind regards, 

on behalf of

Mr. Poowin Bunyavejchewin 

Academic Editor

PLOS ONE